# Well-Controlled Viremia Predicts the Outcome of Hepatocellular Carcinoma in Chronic Viral Hepatitis Patients Treated with Sorafenib

**DOI:** 10.3390/cancers14163971

**Published:** 2022-08-17

**Authors:** Yuan-Hung Kuo, Tzu-Hsin Huang, Jing-Houng Wang, Yen-Yang Chen, Ming-Chao Tsai, Yen-Hao Chen, Sheng-Nan Lu, Tsung-Hui Hu, Chien-Hung Chen, Chao-Hung Hung

**Affiliations:** 1Division of Hepatogastroenterology, Department of Internal Medicine, Kaohsiung Chang Gung Memorial Hospital, Chang Gung University College of Medicine, Kaohsiung 833, Taiwan; 2Division of Hematology-Oncology, Department of Internal Medicine, Kaohsiung Chang Gung Memorial Hospital, Chang Gung University College of Medicine, Kaohsiung 833, Taiwan

**Keywords:** hepatitis B virus, hepatitis C virus, hepatocellular carcinoma, sorafenib, well-controlled viremia

## Abstract

**Simple Summary:**

Previous studies reported hepatitis C virus-related hepatocellular carcinoma (HCV-HCC) patients might have better prognosis than hepatitis B virus-related HCC (HBV-HCC) patients at using sorafenib. However, the information about status of viremia was limited in these studies. We defined well-controlled viremia as patients who had undetectable viremia, or who had been receiving antivirals at least six months before sorafenib. We reported 116 (65.2%) HBV-HCC patients and 62 (34.8%) HCV-HCC patients who received sorafenib, and progression-free survival and overall survival (OS) rates between these two groups were not different. Before sorafenib, 56% of HBV-HCC patients and 54.8% of HCV-HCC patients had well-controlled viremia and their OS was superior to those who had uncontrolled viremia (15.5 vs. 11.1 months, *p* = 0.001). Besides, well-controlled viremia was associated with mortality in multivariate analysis (Hazard ratio: 0.63, 95% confidence interval: 0.42–0.93, *p* = 0.022). The significance of our study is the first research to confirm the prognostic value of well-controlled viremia between HBV-HCC and HCV-HCC patients receiving sorafenib. Besides, HBV or HCV infection was not associated with the outcome, neither in univariate nor in multivariate analysis.

**Abstract:**

Without analyzing the status of viremia, hepatitis C virus-related hepatocellular carcinoma (HCV-HCC) patients are proposed to have better prognosis than hepatitis B virus-related HCC (HBV-HCC) patients using sorafenib. We aimed to elucidate the efficacy of concurrent sorafenib and anti-viral treatment for HCC patients with HBV or HCV infection in real world. Between January 2018 and January 2021, 256 unresectable HCC patients receiving first-line sorafenib were evaluated. High-potency nucleoside analogs were used for HBV control, whereas direct-acting antivirals were administered for HCV eradication. Well-controlled viremia was defined as patients who had undetectable viremia, or who had been receiving antivirals at least 6 months before sorafenib. We recruited 116 (65.2%) HBV-HCC patients and 62 (34.8%) HCV-HCC patients. Using sorafenib, progression-free survival and overall survival (OS) rates between these two groups were not different. Before sorafenib, 56% of HBV-HCC patients and 54.8% of HCV-HCC patients had well-controlled viremia and their OS was superior to those who had uncontrolled viremia (15.5 vs. 11.1 months, *p* = 0.001). Dividing our patients into four subgroups as well-controlled HCV viremia, well-controlled HBV viremia, uncontrolled HCV viremia, and uncontrolled HBV viremia, their OS rates were distributed with a significantly decreasing trend as 21.9 months, 15.0 months, 14.2 months, and 5.7 months (*p* = 0.009). Furthermore, well-controlled viremia was associated with mortality in multivariate analysis (Hazard ratio: 0.63, 95% confidence interval: 0.42–0.93, *p* = 0.022). In real-life, HBV or HCV infection did not contribute to the prognosis of HCC patients receiving sorafenib; however, whether viremia was controlled or not did contribute.

## 1. Introduction

Hepatocellular carcinoma (HCC) is the fourth most common cause of cancer-related death globally and the second leading one in Taiwan [1,2]. Symptomatic HCC patients are usually diagnosed in an advanced stage with major vascular invasion (MVI) or extrahepatic metastasis (EHM), so that most of them are unsuitable for curative treatments and possess an unfavorable prognosis. Sorafenib, an oral multi-kinase inhibitor (MKI), exerts both antiangiogenetic and antitumor effects and has been the first approved target therapy to treat advanced HCC with an unsatisfactory overall survival (OS) rate of less than 1 year [3]. In recent years, with emerging sequential therapies following sorafenib, including new effective target therapies such as regorafenib, cabozantinib, or ramucirumab, as well as immunotherapies such as nivolumab or pembrolizumab and so forth, the OS of patients treated with first-line sorafenib is expected to be further improved [4,5,6,7,8].

In addition, hepatitis B- and C-virus (HBV and HCV) infections represent two of the most common etiologies of HCC, especially in Asia [9]. Concurrent treatment of the underlying viral hepatitis has been demonstrated to ameliorate the outcomes of patients with HBV- or HCV-related HCC [10,11,12,13,14]. For example, high potency nucleotide analogues (NAs) could not only suppress HBV viral load but also reduce the recurrence rate of HBV-related HCC (HBV-HCC), no matter in early or advanced HCC stage [10]. Under the new era of oral direct antiviral agents (DAAs), HCV infection could even be easily eradicated after a short-term treatment course, and this also improves the prognosis of HCV-related HCC (HCV-HCC) [15]. Previous studies have reported that HCV-HCC patients treated with sorafenib seem to have superior median OS over HBV-related HCC (HBV-HCC) patients [16,17]; however, these studies were not head-to-head comparisons. Besides, whether using antiviral drugs or not for the control of viremia was usually not mentioned, but it is real for HBV- or HCV-HCC patients receiving sorafenib. Consequently, the present study attempted to elucidate the efficacy of concurrent sorafenib and anti-viral treatment for unresectable HCC patients with HBV or HCV infection in real-world settings.

## 2. Materials and Methods

### 2.1. Study Cohort

From January 2018 to January 2021, there were 256 consecutive HCC patients in Barcelona clinical liver cancer (BCLC) stages B or C who received sorafenib as the first line systemic therapy in our institute, Kaohsiung Chang Gung Memorial Hospital. All patients received 400 mg of sorafenib twice daily, with the dosage modified by presentation of treatment-related adverse effects (TRAE) based on the manufacturer’s recommendations, while sorafenib monotherapy or in concurrence with other treatment modalities was administered depending on the decision of clinical physicians. After sorafenib termination, post-sorafenib treatments were provided to those patients who still maintained good liver function reserve. Concurrent treatments included liver resection, radiofrequency ablation (RFA), transcatheter arterial chemical embolization (TACE), or radiotherapy, while post-sorafenib treatments included palliative extra-hepatic surgery; TACE; systemic chemotherapy; radiotherapy; hepatic arterial infusion chemotherapy; or previously reported sequential therapies such as regorafenib, lenvatinib, cabozantinib, ramucirumab, nivolumab, pembrolizumab, and so forth [4,5,6,7,8,18]. Patients with HBV-HCC or HCV-HCC were recruited and followed up till December 2021, and their demographics and clinical characteristics were recorded and further analyzed. This current study was approved by the Institutional Review Board of Kaohsiung Chang Gung Memorial Hospital (IRB No: 202200874B0).

### 2.2. Treatment Outcome

The radiologic assessment by liver computer tomography (CT) or magnetic resonance imaging (MRI) was performed based on Response Evaluation Criteria in Solid Tumors version 1.1 every 2 months during sorafenib treatment [19]. Treatment outcomes included OS, meaning the time from treatment initiation to death; progression-free survival (PFS), meaning the time from treatment initiation to disease progression or death; objective response rate (ORR), meaning patients achieved complete response (CR) or partial response (PR); and disease control rate (DCR), meaning patients achieved CR, PR, or stable disease status (SD). TRAE and disease progression were identified from the review of medical records.

### 2.3. Management of Antiviral Therapy

Antiviral therapies with high potency NAs included entecavir, tenofovir disoproxil fumarate (TDF), and tenofovir alafenamide (TAF) and were used for HBV control, whereas peg-interferon-based therapies or pan-genotypic DAAs were administered for HCV eradication. Use of antivirals was according to the Asian-Pacific Association for the Study of the Liver (APASL) clinical practice guidelines [20,21], and all were reimbursed by the National Health Insurance (NHI) of Taiwan. Well-controlled viremia was defined as patients who had undetectable viremia or those who had been receiving antivirals at least 6 months before sorafenib.

### 2.4. Statistical Analysis

All patients were followed up till the last date of visit, death, or the end of December 2021. To compare values between the two groups, chi-squared tests were used to analyze categorical variables, while Student’s *t*-test was applied for continuous variables. Continues variables were expressed with mean ± SD or median with a range; OS and PFS were analyzed using the Kaplan–Meier method with a log-rank test, while univariate and multivariate analyses were performed using Cox proportional hazards regression models. All *p*-values of <0.05 by the two-tailed test were considered significant. All statistical analysis was performed using SPSS 25 software (SPSS Inc., Chicago, IL, USA).

## 3. Results

### 3.1. Clinical Characteristics

Figure 1 shows the flowchart of patient enrollment among 256 unresectable HCC patients under first-line sorafenib treatment. After excluding 6 patients who were lost to follow-up, 10 patients who were infected with HBV plus HCV; 35 with non-HBV, non-HCV; 18 without viremia data; and 9 without etiology records, a total of 178 patients including 116 (65.2%) HBV-HCC and 62 (34.8%) HCV-HCC, were enrolled. Table 1 shows clinical characteristics of all enrolled patients. The mean age was 64.8 years and 75.8% of the patients were male. HCV-HCC patients were older and had a higher proportion of female sex compared with HBV-HCC patients. Based on albumin-bilirubin (ALBI) scoring, 47.2% of patients had ALBI grade I and 52.8% of patients had grade II; additionally, 82% of patients had HCC in BCLC stage C, 50.6% of patients had tumors with MVI, and 43.8% of patients had EHM tumors, respectively. During sorafenib treatment, 25.9% of HBV-HCC patients and 30.6% of HCV-HCC patients also concurrently received other treatment modalities with the two most frequently used treatments being TACE and radiotherapy. Most patients stopped taking sorafenib during the follow-up period. The median duration of sorafenib use was 5.0 months for HBV-HCC patients and 6.2 months for HCV-HCC patients respectively. After sorafenib termination, more HCV-HCC patients were offered post-sorafenib treatments than HBV-HCC patients (75.4% vs. 51.3%, *p* = 0.02). Moreover, regarding second line systemic therapies, the comparison between HCV-HCC and HBV-HCC patients was also significant (61.3% vs. 42.2%, *p* = 0.015). A total of 87 (48.9%) patients received second-line therapies, where regorafenib for 48 patients was the most frequently used agent followed in decreasing order by nivolumab for 21, lenvatinib for 13, other agents for 3, and chemotherapy for 2 patients.

### 3.2. Status of Viremia Control

At the start of sorafenib, comparison of viremia-control between these two groups was not different (56% vs. 54.8%, *p* = 0.878) (Table 1). During sorafenib treatment, a total of 134 (75.3%) patients concurrently received antiviral treatment, and the proportion was higher in HBV-HCC patients compared with HCV-HCC patients (83.6% vs. 59.1%, *p* < 0.001).

### 3.3. Treatment Response

A total of 98 (84.5%) HBV-HCC patients had follow-up dynamic images for the assessment of treatment response (Table 2). Among them, 2% of patients obtained CR, 7.1% achieved PR, 39.8% had SD, and 51.1% became progressive disease (PD). The ORR was 9.1%, whereas the DCR was 48.9%. The duration of sorafenib durability for HBV-HCC was 8.3 months (range: 1.0–24 months). Among 59 (95.2%) HCV-HCC patients with following dynamic images, 5.1% obtained CR, 8.5% achieved PR, 35.6% had SD, and 50.8% became PD, while ORR was 13.6% and DCR was 46.4% respectively. The duration of sorafenib durability for HCV-HCC was 7.9 months (1.0–20.1 months).

### 3.4. Treatment Safety

During sorafenib treatment, HCV-HCC patients had higher proportions of total TRAE than HBV-HCV patients (77.4% vs. 63.8%, *p* = 0.096), but there was no statistical difference (Table 3). In addition, the comparison of severer TRAE (≥grade 3) between HCV-HCC patients and HBV-HCC patients was similar (11.4 vs. 12%). Among HCV-HCC patients, the incidence over 9% included 36.8% of patients with hand–foot skin reaction (HFSR), 22.6% with diarrhea, and 9.4% with fatigue. Six (11.4%) HCV-HCC patients had severe TRAE over grade 3, and three of them were hyperbilirubinemia. Regarding HBV-HCC patients, the top three TRAEs were HFSR (32.4%), diarrhea (25.2%), and decreased appetite (9.6%). Ten (12%) HBV-HCC patients developed grade 3 TRAE requiring treatment termination, and HFSR was the most frequent TRAE (4 of 10).

### 3.5. PFS and OS Based on Virus Hepatitis

Using sorafenib, PFS between HBV-HCC patients and HCV-HCC patients was not different (3.4 months vs. 3.1 months, *p* = 0.619) (Figure 2A). A total of 113 (63.5%) patients died during the follow-up period, including 73 deaths (62.9%) in the HBV-HCC group and 40 deaths (64.5%) in the HCV-HCC group, respectively. OS was 12.4 months in the HBV-HCC patients and 15.5 months in the HCV-HCC patients, respectively (*p* = 0.310) (Figure 2B).

### 3.6. PFS and OS Based on Viremia Control

Based on being viremia-controlled or not, PFS in patients with well-controlled viremia seemed to be longer than those with uncontrolled viremia, but the difference was not significant (4.0 months vs. 2.7 months, *p* = 0.308) (Figure 3A). Concerning the OS, patients with well-controlled viremia were significantly better than those with uncontrolled viremia (15.5 months vs. 11.1 months, *p* = 0.001) (Figure 3B).

### 3.7. PFS and OS Based on Virus Hepatitis plus Viremia Control

Our patients were further divided into four subgroups according to different virus hepatitis and viremia-control status as well-controlled HCV viremia, well-controlled HBV viremia, uncontrolled HCV viremia, and uncontrolled HBV viremia. The distributed PFS were 3.7 months, 4.0 months, 2.7 months, and 2.5 months, respectively (*p* = 0.643). The OS of these four subgroups were distributed with a significant decreasing trend as 21.9 months, 15.0 months, 14.2 months, and 5.7 months (*p* = 0.009). No matter HBV or HCV infection, patients with well-controlled viremia had a significant better OS than those with uncontrolled viremia (15.0 months vs. 5.7 months, *p* = 0.019 in HBV patients (Figure 4A); 21.9 months vs. 14.2 months, *p* = 0.022 in HCV patients (Figure 4B)).

### 3.8. Factors Associated with OS

In multivariate analysis, well-controlled viremia was associated with mortality for patients receiving sorafenib (hazard ratio (HR): 0.63, 95% confidence interval (CI): 0.42–0.93, *p* = 0.022) after adjusting liver function reserve, AFP level, and post-sorafenib treatment (Table 4). HBV or HCV infection, different virus etiologies of HCC, did not contribute to OS, neither for univariate nor multivariate analysis.

## 4. Discussions

In the past decade, sorafenib has been the standard first-line treatment for patients with advanced HCC due to its superior efficacy to other new tyrosine kinase inhibitors and immunotherapeutic agents. Although the combination of atezolizumab (an anti–programmed death-ligand 1 (PD-L1) check-point inhibitor) plus bevacizumab (a vascular endothelial growth factor (VEGF) monoclonal antibody) has been recently proven to have significantly greater survival benefits than sorafenib [22]; high price limits the application of this effective combination. In Taiwan, sorafenib is still the most frequently used first-line agent, because patients with advanced HCC and good liver function reserve could be reimbursed under the National Health Insurance program. In addition, most clinicians are also well-experienced in sorafenib administration so that they could use this agent more effectively and safely in real-life. Moreover, with the development of sequential therapies following sorafenib in recent years, the OS of patients initially receiving sorafenib improved from approximately 10.8 months in the SHARP trial to 14.7 months in the CheckMate 459 trial [3,23]. Consequently, it is still crucial to elucidate which patients are more suitable for sorafenib in real-life.

Preclinical data suggested that sorafenib directly blocked viral replication of HCV so that it could inhibit the inflammatory status related to HCV-HCC [24]. In addition, a subgroup analysis of the SHARP trial showed that HCV-HCC patients treated with sorafenib had relatively superior median OS to HBV-HCC patients (14 months vs. 9.7 months) [16]. An exploratory pooled analysis of another two randomized trials also suggested that sorafenib may be more effective in HCV-HCC [17]. However, all these studies do not clearly provide information about viremia status and whether to use antivirals or not during the period of sorafenib treatment. Indeed, viremia reactivation might induce liver inflammation and deteriorate liver function reserve, which hampers patients in using sorafenib persistently. Previous studies have reported that combining NAs with sorafenib use possessed survival benefits for HBV-HCC patients [13,14]. Antiviral therapy for the management of HBV infection could help reduce hepatic inflammation and preserve liver function during antitumor treatment. Higher risk for reactivation is likely observed in patients who are not on antiviral therapy. Our previous study also indicated that patients using sorafenib concurrently treated with NAs had significantly improved OS compared with patients who received no NAs (8.8 months vs. 4.9 months, *p* = 0.006) [25].

Regarding HCV eradication, current high-potency DAAs could easily achieve sustained virological response (SVR) after a short-term treatment course. However, HCC occurrence was more often observed in HCV-eradicated patients with previous HCC history. A previous study indicated that the occurrence of HCC was 29.6% in patients with previous HCC history, whereas it was only 1.3% in patients without previous HCC history [26]. Thus, for patients with history of HCC, frequent surveillances at the 4-month level are required after HCV eradication. For HCV-HCC patients in early stage, previous studies reported that DAA therapy could improve the survival outcome of HCC patients and did not increase recurrent HCC after curative therapy [15,27]. For HCV-HCC patients in advanced stage under sorafenib treatment, patients receiving DAAs also had a better outcome than patients without [28,29,30]. Kuwano et al. reported that OS was significantly longer in the HCV eradication group than in the HCV non-eradication group (24.0 months vs. 14.1 months; *p* = 0.001) [28]. Seko et al. also indicated that OS in the SVR group were significantly longer than those in the non-SVR group (18.1 months vs. 11.3 months; *p* = 0.019) [29]. Recently, a large database research in Taiwan that enrolled 1,684 HCC patients (122 DAA and 1562 non-DAA users) with first-line sorafenib treatment also indicated that mean survival times were 20.7 months for DAA and 12.5 months for non-DAA (*p* < 0.001) [30]. These real-world studies indicated that no matter whether being HBV-HCC or HCV-HCC patients, those who had well-controlled viremia could obtain better survival outcome than those without; consequently, it is necessary to take the status of viremia into account when clinicians attempt to explore the impact of virus etiologies on HCC patients receiving sorafenib in real-life.

The present study shows that neither PFS nor OS illustrated any difference between HBV-HCC patients and HCV-HCC patients. Although HCV-HCC patients seemed to have longer OS than HBV-HCC patients, the comparison was insignificant (15.5 months vs. 12.4 months, *p* = 0.310). During sorafenib treatment, 83.6% of HBV-HCC patients and 59.1% of HCV-HCC patients concurrently received antiviral treatment. In fact, at the start of sorafenib, there were 56% of HBV-HCC patients and 54.8% of HCV-patients who had received high-potency antivirals at least 6 months previously or had undetectable viremia. We found that patients with well-controlled viremia had significantly superior OS to those with uncontrolled viremia (15.5 months vs. 11.1 months, *p* = 0.001). In fact, patients with well-controlled viremia had a longer sorafenib-use period than patients with uncontrolled viremia, but the comparison was statistically insignificant (6.1months vs. 4.6 months, *p* = 0.063). At the end of sorafenib treatment, the ALBI score in the well-controlled viremia group (−2.20) was significantly lower than that in the uncontrolled viremia group (−1.82) (*p* = 0.002). Hence, patients with well-controlled viremia could be offered more post-treatments than patients with uncontrolled viremia (66.4% vs. 35.6%, *p* = 0.005). Moreover, well-controlled viremia was still associated with mortality in multivariate analysis after adjusting better liver function reserve, lower concentration of AFP, and affording post-treatment. Consequently, when HBV-HCC patients or HCV-HCC patients attempt to start sorafenib, it is necessary to use antivirals concurrently for those who still have detectable viremia.

We further divided our patients into four subgroups according to different virus hepatitis and viremia-control status as well-controlled HCV viremia, well-controlled HBV viremia, uncontrolled HCV viremia and uncontrolled HBV viremia, and the OS of these four subgroups were distributed with a significantly decreasing trend as 21.9 months, 15.0 months, 14.2 months, and 5.7 months (*p* = 0.009). Same as previous studies, the well-controlled HCV group had longer survival time than the uncontrolled HCV group (21.9 months vs. 14.2 months, *p* = 0.022), whereas the well-controlled HBV group also had better survival outcome than the uncontrolled HBV group (15.0 months vs. 5.7 months, *p* = 0.019). Unlike HBV viremia suppression induced by anti-HBV NAs, current pan genotypic DAAs could really eradicate HCV and possibly lead to better virus control. Our well-controlled HCV group also seemed to have superior OS to the well-controlled HBV group, but the comparison was insignificant (21.9 months vs. 15.0 months, *p* = 0.298). Relatively smaller sample size and shorter follow-up period might partially explain why there was no statistical OS difference between these two groups.

Post-treatment also contributed to better prognosis in the present study. Patients who could receive post-treatment had significantly superior OS than those who could not (20.1 months vs. 4.9 months, *p* < 0.001). After sorafenib termination, more HCV-HCC patients maintained good liver function reserve and afforded post-sorafenib treatments (75.4% vs. 51.3%, *p* = 0.02) and second-line systemic therapies (61.3% vs. 42.2%, *p* = 0.015) than HBV-HCC patients. A total of 87 (48.9%) patients received sequential therapies, where regorafenib for 48 patients was the most frequently used agent followed in decreasing order by nivolumab, lenvatinib, and chemotherapy.

The present study still has some limitations. Firstly, this was a retrospective study so that some patients lacked viremia data at the time sorafenib administration commenced. In addition, some patients also lacked follow-up dynamic images for the assessment of treatment response due to deterioration of performance status, death, or unrecorded reasons. Secondly, in clinical real practice, baseline characteristics of HBV-HCC and HCV-HCC groups including age, sex, and platelet count were not consistent, which might lead to confounding bias in the analysis. Thirdly, the well-controlled HCV-HCC group seemed to have longer OS than the well-controlled HBV-HCC group, but the comparison was not significant. Further large sample-sized studies are required to reduce these possible statistical biases.

## 5. Conclusions

In clinical real-life, HBV or HCV infection did not contribute to the prognosis of HCC patients receiving sorafenib; however, being viremia-controlled or not did contribute. Consequently, when HBV-HCC or HCV-HCC patients attempt to start sorafenib, it is necessary to use antivirals concurrently for those who still have detectable viremia.

## Figures and Tables

**Figure 1 cancers-14-03971-f001:**
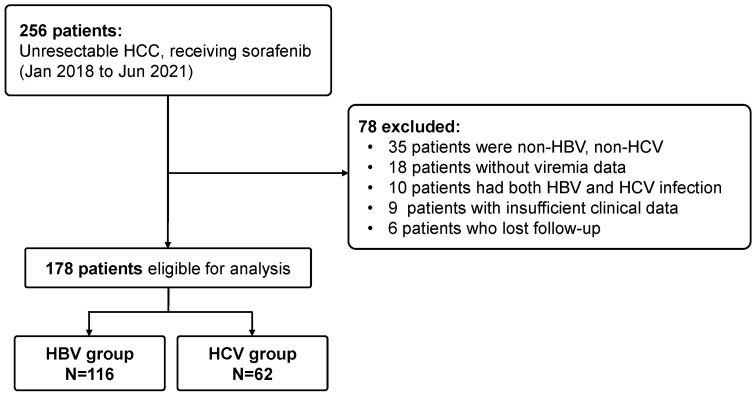
Flow chart of all enrolled patients.

**Figure 2 cancers-14-03971-f002:**
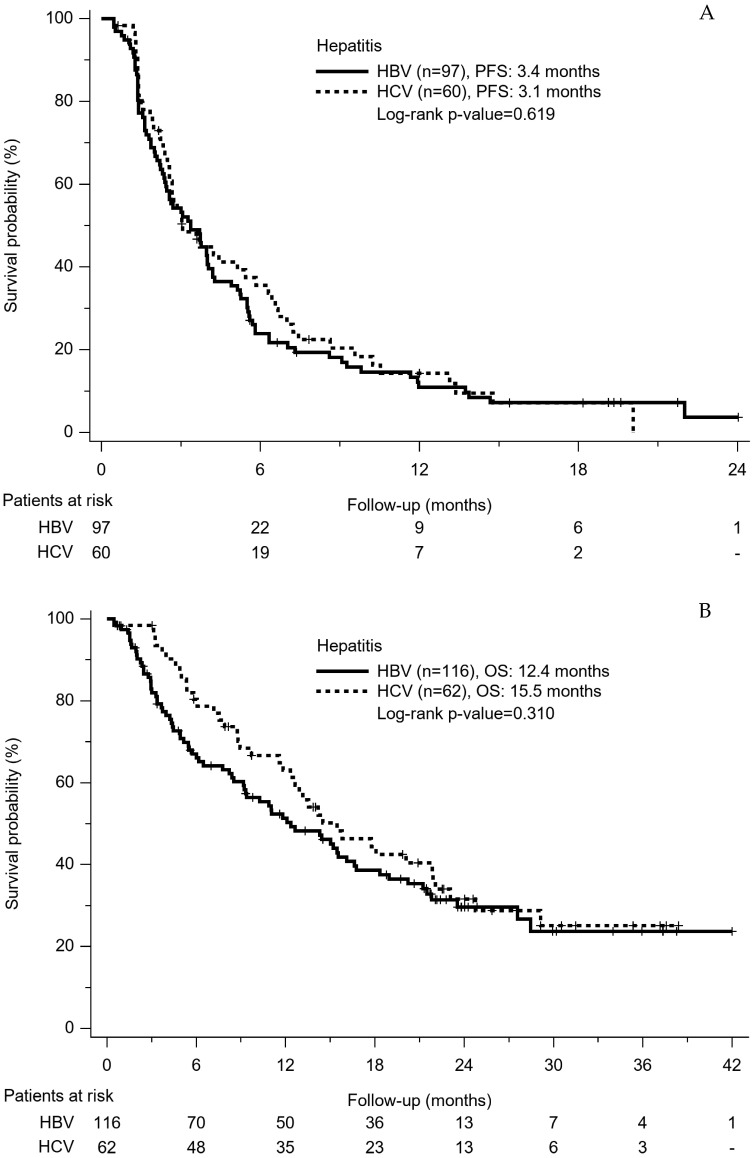
(**A**) Progression-free survival rates of HBV-HCC patients and HCV-HCC patients. (**B**) Overall survival rates of HBV-HCC patients and HCV-HCC patients.

**Figure 3 cancers-14-03971-f003:**
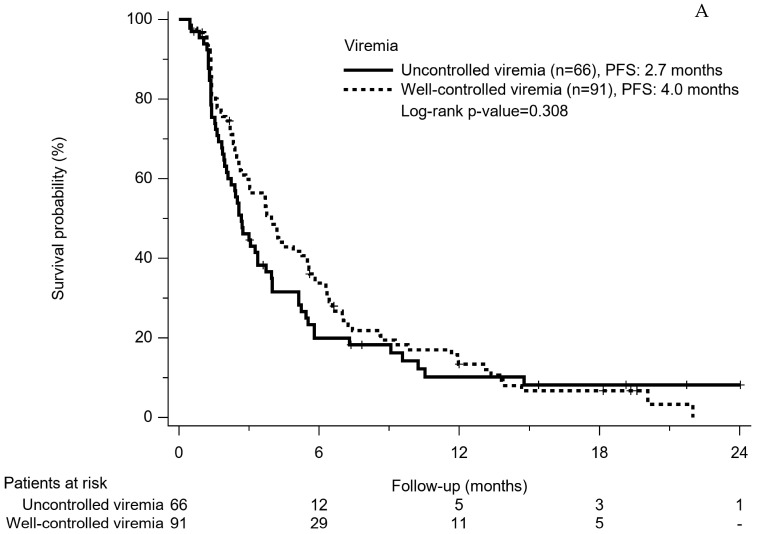
(**A**). Progression-free survival rates of patients with well-controlled viremia and uncontrolled viremia. (**B**). Overall survival rates of patients with well-controlled viremia and uncontrolled viremia.

**Figure 4 cancers-14-03971-f004:**
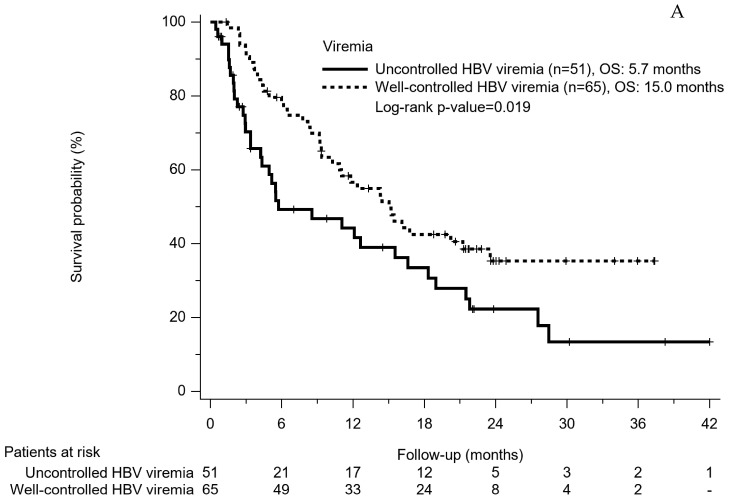
(**A**). Overall survival rates based on HBV infection and viremia-control status. (**B**) Overall survival rates based on HCV infection and viremia-control status.

**Table 1 cancers-14-03971-t001:** Demographics and clinical characteristics of enrolled HBV-HCC patients and HCV-HCC patients.

	Total (*n* = 178)	HBV-HCC (*n* = 116)	HCV-HCC (*n* = 62)	*p*-Value
Follow-up interval, months	13.4 ± 10.2	12.3 ± 10.1	15.3 ± 9.9	0.061
Age(years)	64.8 ± 11.4	61.7 ± 11.6	70.5 ± 8.7	<0.001
Male sex, *n* (%)	135 (75.8)	93 (80.2)	42 (67.7)	0.065
Child-Pugh class, A, *n* (%)	174 (97.8)	115 (99.1)	59 (95.2)	0.088
B, *n* (%)	4 (2.2)	1 (0.9)	3 (4.8)	
ALBI grade 1, *n* (%)	84 (47.2)	53 (45.7)	31 (50)	0.563
2, *n* (%)	94 (52.8)	63 (54.3)	31 (50)	
BCLC stage, B, *n* (%)	32 (18)	18 (15.5)	14 (22.6)	0.242
C, *n* (%)	146 (82)	98 (84.5)	48 (77.4)	
EHM, *n* (%)	78 (43.8)	55 (47.4)	23 (37.1)	0.186
MVI, *n* (%)	90 (50.6)	60 (51.7)	30 (48.4)	0.671
Tumor size ≥ 6 cm, *n* (%)	57 (41.9)	42 (46.7)	15 (32.6)	0.116
AFP, ng/mL	7701 ± 2022	8714 ± 2172	5774 ± 2077	0.333
AFP ≥ 200 ng/mL, *n* (%)	86 (48.8)	62 (53.4)	24 (39.3)	0.074
AST, IU/L	69.6 ± 58.7	72.3 ± 63.3	64.3 ± 48.7	0.354
ALT, IU/L	54.5 ± 53.7	56.8 ± 59.9	50.0 ± 39.2	0.367
Total Bilirubin, mg/dL	1.0 ± 0.5	1.1 ± 0.5	1.0 ± 0.5	0.122
Albumin, g/dL	3.9 ± 0.5	3.9 ± 0.6	3.9 ± 0.5	0.9
Platelet, ×10^3^ /uL	161 ± 101	174 ± 106	136 ± 87	0.058
PT INR	1.06 ± 0.1	1.06 ± 0.1	1.06 ± 0.1	0.594
Concurrent treatment, *n* (%)	49 (27.5)	30 (25.9)	19 (30.6)	0.496
Post treatment, *n* (%)	104 (59.8)	58 (51.3)	46 (75.4)	0.02
Second line systemic treatment, *n* (%)	87 (48.9)	49 (42.2)	38 (61.3)	0.015
Regorafenib, *n*	48	23	25	
Nivolumab, *n*	21	13	8	
Lenvatinib, *n*	13	8	5	
Chemotherapy, *n*	2	2	0	
Others, *n*	3	3	0	
Antiviral treatment, *n* (%)	134 (75.3)	97 (83.6)	37 (59.1)	<0.001
Viremia at sorafenib-start, log IU/mL		4.5 ± 1.7	5.6 ± 0.9	<0.001
Well-controlled viremia, *n* (%)	99 (55.6)	65 (56)	34 (54.8)	0.878
Sorafenib stop, *n* (%)	174 (97.8)	113 (97.4)	61 (98.4)	0.676
Sorafenib-use, months	5.4 ± 5.3	5.0 ± 5.3	6.2 ± 5.2	0.68

Abbreviation: AFP, alpha-fetoprotein; ALBI grade, albumin-bilirubin grade; ALT, alanine aminotransferase; AST, aspartate transaminase; EHM: extra-hepatic metastasis; HBV, hepatitis B virus; HBV-HCC, HBV-related HCC; HCC, hepatocellular carcinoma; HCV, hepatitis C virus; HCV-HCC, HCV-related HCC; MVI, macro-vascular invasion; N: number; PT INR, prothrombin time international normalized ratio.

**Table 2 cancers-14-03971-t002:** Treatment response of HCC patients with sorafenib.

Variables	HBV-HCC (*n* = 116)	HCV-HCC (*n* = 62)
Treatment response evaluation, *n* (%)	98 (84.5)	59 (95.2)
Complete response	2 (2.0)	3 (5.1)
Partial response	7 (7.1)	5 (8.5)
Stable disease	39 (39.8)	21 (35.6)
Progression disease	50 (51.1)	30 (50.8)
Objective response rate, %	9.1	13.6
Disease control rate, %	48.9	46.4
Durability of response, month	8.3 (1.0–24)	7.9 (1.0–20.1)
Death, *n* (%)	73 (62.9)	40 (64.5)

Abbreviation: HBV, hepatitis B virus; HBV-HCC, HBV-related HCC; HCC, hepatocellular carcinoma; HCV, hepatitis C virus; HCV-HCC, HCV-related HCC; N: number.

**Table 3 cancers-14-03971-t003:** Treatment-related adverse events (TRAE) in HBV-HCC and HCV-HCC patients.

	HBV-HCC (*n* = 83) *		HCV-HCC (*n* = 53) *	
	Any, *n* (%)	Grade ≥ 3, *n* (%)	Any, *n* (%)	Grade ≥ 3, *n* (%)
Total patients with TRAE *n* (%)	53 (63.8)	10 (12)	41 (77.4)	6 (11.4)
Hand foot skin reaction, *n* (%)	27 (32.4)	4 (4.8)	20 (36.8)	2 (3.8)
Diarrhea, *n* (%)	21 (25.2)	1 (1.2)	12 (22.6)	1 (1.9)
Decreased appetite, *n* (%)	8 (9.6)	0	1 (1.9)	0
Fatigue, *n* (%)	3 (3.6)	2 (2.4)	5 (9.4)	0
Increased AST, *n* (%)	3 (3.6)	2 (2.4)	0	0
Dermatitis, *n* (%)	3 (3.6)	0	3 (5.7)	0
Pruritus, *n* (%)	2 (2.4)	0	0	0
Increased T-bil, *n* (%)	2 (2.4)	1 (1.2)	3 (5.7)	3 (5.7)
Hypertension, *n* (%)	1 (1.2)	0	3 (5.7)	0
Hypothyroidism, *n* (%)	1 (1.2)	0	0	0

Abbreviation: AST, aspartate transaminase; HBV, hepatitis B virus; HBV-HCC, HBV-related HCC; HCC, hepatocellular carcinoma; HCV, hepatitis C virus; HCV-HCC, HCV-related HCC; N: number; T-bil, total bilirubin. * Comparison of treatment-related adverse events was based on those patients who had medical records. The comparison of any TRAE between two groups was 0.096.

**Table 4 cancers-14-03971-t004:** Factors associated with overall survival of patients with sorafenib in the univariate and multivariate analysis.

		Univariate Analysis			Multivariate Analysis		
Variable	Comparison	H.R.	95% CI	*p*-value	H.R.	95% CI	*p*-value
Age, years	Increase per year	0.99	0.98–1.01	0.439	1.01	0.99–1.03	0.592
Sex	Female vs. Male	1.10	0.72–1.69	0.65	0.97	0.61–1.55	0.911
ALBI-grade	II vs. I	2.05	1.40–2.99	<0.001	1.80	1.19–2.73	0.006
BCLC stage	C vs. B	1.73	1.02–2.94	0.043	1.89	0.82–4.32	0.135
EHM	Yes vs. No	1.13	0.78–1.64	0.52	1.06	0.58–1.91	0.856
MVI	Yes vs. No	1.49	1.03–2.16	0.035	0.95	0.51–1.79	0.872
AFP ≥ 200 ng/ml	Yes vs. No	2.09	1.43–3.04	<0.001	1.66	1.08–2.54	0.02
Concurrent treatment	Yes vs. No	1.04	0.70–1.55	0.858	0.89	0.58–1.34	0.568
Post treatment	Yes vs. No	0.27	0.18–0.40	<0.001	0.34	0.22–0.53	<0.001
HCC Etiology	HCV vs. HBV	0.82	0.56–1.21	0.311	0.92	0.59	1.43
Well-controlled viremia	Yes vs. No	0.55	0.38–0.80	0.002	0.63	0.42–0.93	0.022

Abbreviation: AFP, alpha-fetoprotein; ALBI grade, albumin-bilirubin grade; BCLC stage, Barcelona clinical liver cancer stage; CI, confidence interval; EHM: extra-hepatic metastasis; HBV, hepatitis B virus; HCC, hepatocellular carcinoma; HCV, hepatitis C virus; HR, Hazard ratio; MVI, macro-vascular invasion.

## Data Availability

Data are contained within the article.

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
