# Peer review of "Well-Controlled Viremia Predicts the Outcome of Hepatocellular Carcinoma in Chronic Viral Hepatitis Patients Treated with Sorafenib"

_cancers, 2022, doi:10.3390/cancers14163971_

Round 1

Reviewer 1 Report

The retrospective study by Kuo et al focused on investigating the association between viremia control and therapeutic efficacy of sorafenib, a clinical in-used TKI inhibitor for advanced HCC patients. By grouping HCC patients into HBV- or HCV-related HCC, the analysis on 196 patients indicated that the type of hepatitis virus may not be related to the therapeutic efficacy of sorafenib, represented by both PFS and OS. Interestingly, patients with well-controlled viremia, regardless of HBV or HCV, showed a better prognosis under sorafenib treatment. Although other well-known factors, including cancer stage, liver function, and sequential and continuous treatment of other drugs contributed to patient prognosis, it is important to point out the potential role of virus status in the therapeutic efficacy of sorafenib. Overall, the study is informative and of potential clinical impact.   

Major points:

1. The patient information from Fig. 1 listed a group of 35 non-B no-C patients with HCC who were also under the sorafenib treatment. Since the study indicated that viremia control is associated with sorafenib, it is also interesting to compare the patient prognosis of non-B non-C, well-controlled viremia, towards uncontrolled ones. The results will further emphasize the importance of virus status in sorafenib treatment.

2. The author defined well-controlled viremia as ‘undetectable viremia’ and ‘patients who had been receiving antivirals at least three months before sorafenib’. If the virus level/status really matters, this definition may not be convincing. Clear criteria of virus level for defining ‘well-controlled viremia’ may help.    

Author Response

The retrospective study by Kuo et al focused on investigating the association between viremia control and therapeutic efficacy of sorafenib, a clinical in-used TKI inhibitor for advanced HCC patients. By grouping HCC patients into HBV- or HCV-related HCC, the analysis on 196 patients indicated that the type of hepatitis virus may not be related to the therapeutic efficacy of sorafenib, represented by both PFS and OS. Interestingly, patients with well-controlled viremia, regardless of HBV or HCV, showed a better prognosis under sorafenib treatment. Although other well-known factors, including cancer stage, liver function, and sequential and continuous treatment of other drugs contributed to patient prognosis, it is important to point out the potential role of virus status in the therapeutic efficacy of sorafenib. Overall, the study is informative and of potential clinical impact.   

Major points:

  1. The patient information from Fig. 1 listed a group of 35 non-B no-C patients with HCC who were also under the sorafenib treatment. Since the study indicated that viremia control is associated with sorafenib, it is also interesting to compare the patient prognosis of non-B non-C, well-controlled viremia, towards uncontrolled ones. The results will further emphasize the importance of virus status in sorafenib treatment.

Reply: We appreciated the reviewer`s recommendations and we have been analyzed the OS among these groups before. However, the OS among patients with HBV, HCV, non-B non-C were 12.4 months, 15.5 months, and 9.8 months, respectively (p=0.335), whereas the OS among patients with well-controlled viremia, uncontrolled viremia, and non-B non-C were15.4 months, 11.1 months, and 9.8 months, respectively (p=0.005). Non-B non-C patients seemed to have the worst outcome comparing with viral hepatitis patients with or without viremia-control. Different tumor behavior, lack of regular HCC surveillance or coexistence of other etiologies such as alcoholism, obesity, diabetes etc. might partially explain why our non-viral patients had the worst prognosis than other viral hepatitis patients. Maybe we could conduct another study to elucidate the difference of OS between viral-HCC patients and non-viral HCC patients in the future.

  1. The author defined well-controlled viremia as ‘undetectable viremia’ and ‘patients who had been receiving antivirals at least three months before sorafenib’. If the virus level/status really matters, this definition may not be convincing. Clear criteria of virus level for defining ‘well-controlled viremia’ may help.    

Reply: We appreciated the reviewer`s recommendations and we have attempted to make our definition of “well-controlled viremia” clearer. Because the published guidelines by WHO suggests that treatment of chronic HBV usually takes 24-48 weeks to reach low or undetectable HBV viral loads. And SVR is defined as undetectable HCV viral loads at 12 weeks later after a complete 8-weeks or 12-weeks DAA course. Hence, well-controlled viremia may be more appropriate to define as patients who had undetectable viremia or those who had been receiving antivirals at least six months before sorafenib. After rechecking our study cohort, we excluded 18 patients without clear viremia data in six months before sorafenib treatment. Certainly, those HBV patients have been receiving NAs and had undetectable viremia data, or HCV patients had achieved SVR by DAAs more than six months before sorafenib treatment still be enrolled in this study.

Reviewer 2 Report

The authors attempted to determine the efficacy of concurrent sorafenib and anti-viral treatment for unresectable HCC patients with HBV or HCV infection in real-world settings in this study. Overall, the manuscript appears to be excellent and may provide significant knowledge to the scientific community. I strongly recommend that the manuscript be accepted in its current form.

Author Response

The authors attempted to determine the efficacy of concurrent sorafenib and anti-viral treatment for unresectable HCC patients with HBV or HCV infection in real-world settings in this study. Overall, the manuscript appears to be excellent and may provide significant knowledge to the scientific community. I strongly recommend that the manuscript be accepted in its current form.

Reply: We appreciated the reviewer`s recommendations.

Reviewer 3 Report

The difference in OS e PSF between HBV and HCV patients with advanced HCC treated with systemic therapy has been largely described. Your focus on controlled/uncontrolled viremia is very interesting.

I have only few suggestions/remarks:

1- In M&M you said "Concurrent treatments included liver resection, radiofrequency ablation (RFA), transcatheter arterial chemical embolization (TACE), radiotherapy or nivolumab..."
Do you have enrolled patients treated with first-line sorafenib+nivolumab combination therapy? If yes, I think that this cohort of patients should be excluded because immunotherapy could get OS and PFS longer;

2- In M&M you said "Well-controlled viremia was defined as patients who had undetectable viremia or those who had been receiving antivirals at least three months before sorafenib."
I'm not sure I have understand. Have you considered as "well-contolled" patients on antiviral treatment for at least 3 months even if viremia was detectable? If no, you have to better explain this; if yes, do you have considered any breakpoint of viremia? Being on treatment is not sufficient for consider a patient as "well-controlled";

3- In results (paragraph 2.3) you said "A total of 100 (80.6%) HBV-HCC patients had follow-up dynamic images for the assessment of treatment response (Table 2)......Among 65 (90.3%) HCV-HCC patients with following dynamic images...".
In the flow chart for enrollment you correctly excluded patients who lost follow up, so, I suppose that the 19.4% of HBV-HCC and 9.7% of HCV-HCC patients that do not have a follow up imaging is for other reasons. Which one? Are they died before the first radiologic re-evaluation? If yes, they have to be considered "PD" as best treatment response and, consequently, DCR and ORR re-calculated. Anyway, this paragraph need more explanations;

4- Figure 2A, 3A, 3B, 4A, 4B: Why have you considered for analysis only a part of patients? Your study populations is 196 patients (124 HBV + 72 HCV), but in the above-mentioned Kaplan-Meier curves you have considered only 172, 157, 179, 158 and 180 patients, respectively.

5- Table 4: BCLC and MVI were statistically significant in univariate analysis.  I suppose that these two variables were considered in the multivariate analysis and were statistically unsignificant. Anyway, it is more correct to provide HR, 95CI and p-value.

Author Response

The difference in OS e PSF between HBV and HCV patients with advanced HCC treated with systemic therapy has been largely described. Your focus on controlled/uncontrolled viremia is very interesting.

I have only few suggestions/remarks:

1- In M&M you said "Concurrent treatments included liver resection, radiofrequency ablation (RFA), transcatheter arterial chemical embolization (TACE), radiotherapy or nivolumab..."
Do you have enrolled patients treated with first-line sorafenib+nivolumab combination therapy? If yes, I think that this cohort of patients should be excluded because immunotherapy could get OS and PFS longer;

Reply: We appreciated the reviewer`s recommendations and the two patients treated with sorafenib plus nivolumab has been also excluded due to no viremia data before sorafenib treatment.

2- In M&M you said "Well-controlled viremia was defined as patients who had undetectable viremia or those who had been receiving antivirals at least three months before sorafenib."
I'm not sure I have understand. Have you considered as "well-contolled" patients on antiviral treatment for at least 3 months even if viremia was detectable? If no, you have to better explain this; if yes, do you have considered any breakpoint of viremia? Being on treatment is not sufficient for consider a patient as "well-controlled";

Reply: We appreciated the reviewer`s recommendations and we have attempted to make our definition of “well-controlled viremia” clearer. Because the published guidelines by WHO suggests that treatment of chronic HBV usually takes 24-48 weeks to reach low or undetectable HBV viral loads. And SVR is defined as undetectable HCV viral loads at 12 weeks later after a complete 8-weeks or 12-weeks DAA course. Hence, well-controlled viremia may be more appropriate to define as patients who had undetectable viremia or those who had been receiving antivirals at least six months before sorafenib. After rechecking our study cohort, we excluded 18 patients without clear viremia data in six months before sorafenib treatment. Certainly, those HBV patients have been receiving NAs and had undetectable viremia data, or HCV patients had achieved SVR by DAAs more than six months before sorafenib treatment still be enrolled in this study.

3- In results (paragraph 2.3) you said "A total of 100 (80.6%) HBV-HCC patients had follow-up dynamic images for the assessment of treatment response (Table 2)......Among 65 (90.3%) HCV-HCC patients with following dynamic images...".
In the flow chart for enrollment you correctly excluded patients who lost follow up, so, I suppose that the 19.4% of HBV-HCC and 9.7% of HCV-HCC patients that do not have a follow up imaging is for other reasons. Which one? Are they died before the first radiologic re-evaluation? If yes, they have to be considered "PD" as best treatment response and, consequently, DCR and ORR re-calculated. Anyway, this paragraph need more explanations;

Reply: We appreciated the reviewer`s recommendations. After rechecking our study cohort, we excluded 18 patients without clear viremia data in six months before sorafenib treatment. The recalculated data showed that 18 (15.5%) HBV-HCC patients and 3 (4.8%) HCV-HCC patients lacked the follow-up dynamic images. Some patients didn`t receive the image study due to deterioration of performance status, death, or unrecorded reasons. And we have described it as one of our study limitations as “Firstly, this was a retrospective study so that some patients lacked viremia data at the time sorafenib administration commenced. And some patients also lacked follow-up dynamic images for the assessment of treatment response due to deterioration of performance status, death, or unrecorded reasons.” In line 18, page 17 in the revised text.

4- Figure 2A, 3A, 3B, 4A, 4B: Why have you considered for analysis only a part of patients? Your study populations is 196 patients (124 HBV + 72 HCV), but in the above-mentioned Kaplan-Meier curves you have considered only 172, 157, 179, 158 and 180 patients, respectively.

Reply: We appreciated the reviewer`s recommendations. After rechecking our study cohort, we excluded 18 patients without clear viremia data in six months before sorafenib treatment. Patient numbers of the OS of all Kaplan-Meier curves were 178 patients. Patient numbers of the PFS of Kaplan-Meier curves were based on the patients with clear image response or death identification.

5- Table 4: BCLC and MVI were statistically significant in univariate analysis.  I suppose that these two variables were considered in the multivariate analysis and were statistically unsignificant. Anyway, it is more correct to provide HR, 95CI and p-value.

Reply: We appreciated the reviewer`s recommendations. Indeed, these two variables were significant in univariate analysis but insignificant in multivariate analysis. And we have provided HR, 95CI and p-value of all variables according to the reviewer`s recommendations in the revised table.

Reviewer 4 Report

Comments: 

The study submitted by Kuo Y-H et al. provided valuable insight and evidence to the utility of anti-viral HBV or HCV therapies even in patients with unresectable HCC. Overall, they provide a well-written and interesting contribution towards the management of patients with HBV-HCC and HCV-HCC. However, there are some concerns with the current version of the manuscript that needs clarification or revision: 

1. What is the evidence of defining well-controlled viremia as those using anti-virals for 3 months before sorafenib? In most studies and clinical guidelines for chronic HBV infection, well-controlled viremia is usually defined with low serum viral loads (eg. <2000 IU/mL) for many months with regular intervals (usually 3 months) of monitoring. Even the published guidelines by WHO suggests that treatment of chronic HBV usually takes 24-48 weeks (>3 months) to reach low or undetectable HBV viral loads. The definition used by the authors is insufficient to be appropriately called well-controlled disease at least in the context of chronic HBV. A more stringent definition of well-controlled HBV could impact the findings and comparisons between those well-controlled and uncontrolled. 

2. A suggestion would be to separate the viral hepatitis of Figures 4A and B as it is currently quite difficult to read and interpret. One set of figures to compare outcomes of uncontrolled vs. well-controlled HBV and a second set of figures to compare HCV would be clearer than the current Figures 4A and B. 

3. Why did patients stop taking sorafenib? More specifically, did any patients stop sorafenib due to tolerance/adverse effects? Are there any significant differences in patients either HBV vs. HCV or well-controlled vs. uncontrolled viremia and their rates of sorafenib cessation?  

4. There are no explanations or section to describe the findings of Table 3. 

5. Additional potential findings of interest could be treatment responses to sorafenib, time to stoppage of sorafenib, transition to second-line systemic therapy, etc. in the context of well-controlled vs. uncontrolled subgroups. For example, could those who have well-controlled viremia have better and longer treatment responses with sorafenib (ie. longer time to cessation of sorafenib?).

Additionally, there are some more specific but minor comments: 

Minor typographical errors: 

• Page 2 - 2.2 Treatment outcome

o “progression-free survival”

Please consistently clarify all acronyms when first introduced: 

• Page 2 - 2.2 Treatment outcome

o “ORR”

o “DCR”

• Page 3 – 3.1 Clinical characteristics: 

o “RTO”

Author Response

Comments: 

The study submitted by Kuo Y-H et al. provided valuable insight and evidence to the utility of anti-viral HBV or HCV therapies even in patients with unresectable HCC. Overall, they provide a well-written and interesting contribution towards the management of patients with HBV-HCC and HCV-HCC. However, there are some concerns with the current version of the manuscript that needs clarification or revision: 

  1. What is the evidence of defining well-controlled viremia as those using anti-virals for 3 months before sorafenib? In most studies and clinical guidelines for chronic HBV infection, well-controlled viremia is usually defined with low serum viral loads (eg. <2000 IU/mL) for many months with regular intervals (usually 3 months) of monitoring. Even the published guidelines by WHO suggests that treatment of chronic HBV usually takes 24-48 weeks (>3 months) to reach low or undetectable HBV viral loads. The definition used by the authors is insufficient to be appropriately called well-controlled disease at least in the context of chronic HBV. A more stringent definition of well-controlled HBV could impact the findings and comparisons between those well-controlled and uncontrolled. 

Reply: We appreciated the reviewer`s recommendations and we have attempted to make our definition of “well-controlled viremia” clearer. Because the published guidelines by WHO suggests that treatment of chronic HBV usually takes 24-48 weeks to reach low or undetectable HBV viral loads. And SVR is defined as undetectable HCV viral loads at 12 weeks later after a complete 8-weeks or 12-weeks DAA course. Hence, well-controlled viremia may be more appropriate to define as patients who had undetectable viremia or those who had been receiving antivirals at least six months before sorafenib. After rechecking our study cohort, we excluded 18 patients without clear viremia data in six months before sorafenib treatment. Certainly, those HBV patients have been receiving NAs and had undetectable viremia data, or HCV patients had achieved SVR by DAAs more than six months before sorafenib treatment still be enrolled in this study.

  1. A suggestion would be to separate the viral hepatitis of Figures 4A and B as it is currently quite difficult to read and interpret. One set of figures to compare outcomes of uncontrolled vs. well-controlled HBV and a second set of figures to compare HCV would be clearer than the current Figures 4A and B. 

Reply: We appreciated the reviewer`s recommendations and we have separated different viral hepatitis. One set of figures to compare outcomes of uncontrolled vs. well-controlled HBV and a second set of figures to compare HCV in the revised text.

  1. Why did patients stop taking sorafenib? More specifically, did any patients stop sorafenib due to tolerance/adverse effects? Are there any significant differences in patients either HBV vs. HCV or well-controlled vs. uncontrolled viremia and their rates of sorafenib cessation?  

Reply: We appreciated the reviewer`s recommendations. There were 10 (8.6%) HBV-HCC patients and 6 (9.7%) HCV-HCC patients stopped sorafenib due to treatment related adverse effect (TRAE). Based on viremia control, there were 9 (9.1%) well-controlled viremia patients and 7 (8.9%) uncontrolled viremia patients stopped sorafenib due to TRAE. And the comparisons of sorafenib cessation by TRAE between these patients were insignificant.

  1. There are no explanations or section to describe the findings of Table 3. 

Reply: We appreciated the reviewer`s reminder and we have described the finding of

table 3 as During sorafenib treatment, HCV-HCC patients had higher proportions of total

TRAE than HBV-HCV patients (77.4% vs 63.8%, p=0.096), but there was no statistical difference (Table 3). In addition, the comparison of severer TRAE (≥ grade 3) between HCV-HCC patients and HBV-HCC patients was similar (11.4 vs 12%). Among HCV-HCC patients, the incidence over 9% included 36.8% of patients with hand-foot skin reaction (HFSR), 22.6% with diarrhea, and 9.4% with fatigue. Six (11.4%) HCV-HCC patients had severe TRAE over grade 3, and three of them were hyperbilirubinemia. Regarding HBV-HCC patients, the top three TRAEs were HFSR (32.4%), diarrhea (25.2%), and decreased appetite (9.6%). Ten (12%) HBV-HCC patients developed grade 3 TRAE requiring treatment termination, and HFSR was the most frequent TRAE (4 of 10). In line 8, page 10 in the revised text.

  1. Additional potential findings of interest could be treatment responses to sorafenib, time to stoppage of sorafenib, transition to second-line systemic therapy, etc. in the context of well-controlled vs. uncontrolled subgroups. For example, could those who have well-controlled viremia have better and longer treatment responses with sorafenib (ie. longer time to cessation of sorafenib?).

Reply: We appreciated the reviewer`s recommendations and we have discussed this issue as “In fact, patients with well-controlled viremia had longer sorafenib-use period than patients with uncontrolled viremia, but the comparison was statistically insignificant (6.1months vs 4.6 months, p=0.063). At the end of sorafenib treatment, the ALBI score in the well-controlled viremia group (- 2.20) was significantly lower than that in the uncontrolled viremia group (- 1.82) (p = 0.002). Hence, patients with well-controlled viremia could be offered more post-treatments than patients with uncontrolled viremia (66.4% vs 35.6%, p=0.005).” in line 2, page 16 in the revised  text.

Additionally, there are some more specific but minor comments: 

Minor typographical errors: 

  • Page 2 - 2.2 Treatment outcome

o “progression-free survival”

Reply: We appreciated the reviewer`s reminder and we have corrected this error.

Please consistently clarify all acronyms when first introduced: 

  • Page 2 - 2.2 Treatment outcome

o “ORR”

o “DCR”

  • Page 3 – 3.1 Clinical characteristics: 

o “RTO”

 Reply: We appreciated the reviewer`s reminder and we have corrected these errors.
